# Exploring Exploration: A Comparative Analysis of Colored Noise Strategies in Reinforcement Learning

## Abstract

Reinforcement Learning algorithms, in general, and off-policy agents navigating continuous control spaces, in particular, often induce exploration through the addition of noise into their action selection process. Popular implementations majorly utilize uncorrelated Gaussian (white) noise, or temporally correlated Ornstein-Uhlenbeck (OU) noise, which is closely related to red noise. Eberhard et al. (2023) propose using pink noise, which is halfway between white and OU noise, as the default action noise type. The authors also claim pink noise to be a better default than noise schedulers, which are algorithms that vary the level of temporal correlation as learning progresses. In this paper, we attempt to verify their claims and present an analysis of colored noise exploration, comparing various strategies of noise integration. We further attempt to identify the effect of using spatially and temporally correlated noise to achieve exploration. The code and samples are present in the supplementary material.

## 1 Introduction

In recent years, deep reinforcement learning (DRL) has emerged as a powerful paradigm at the intersection of artificial intelligence and machine learning, revolutionizing the way autonomous agents learn to interact with complex environments. One major challenge faced by most algorithms is exploration. Further complications arise when dealing with continuous control environments, as agents must exhibit a sequence of precise actions to reach a sufficiently distinct state.

Achieving a high-performing policy hinges on gathering data (i.e., state-action-reward sequences) of sufficiently diverse behaviors. Most off-policy reinforcement learning algorithms utilize the addition of stochastic action noise to the action chosen by the policy. This effectively models exploration and traditionally, this noise takes the form of either uncorrelated white noise or temporally correlated Ornstein-Uhlenbeck (OU) noise (Uhlenbeck & Ornstein, 1930).

Both white and OU noise have inherent limitations, which result in poor exploration in certain environments. White noise, lacking temporal correlation, may fail to explore distant, high-reward states adequately. OU noise has the opposite problem. It favors global exploration and can lead to extremely off-policy trajectories. This makes it difficult for the model to learn the on-policy state visitation distribution.

Eberhard et al. (2023) propose an intermediate to both of these noise types, Pink Noise. In theory, pink noise achieves a median between both local and global exploration, and the authors claim it to be a better default than the commonly used noise types. We aim to reproduce the results claimed by the authors in this report.

To deal with the problems of White and OU noise described above, another method is the usage of noise schedulers. Typical schedulers decay the noise from more temporally correlated noise types to lesser correlated ones. This allows for global exploration in the beginning, when the model has not learned anything, and local exploration later when the model has sufficient learnings. In addition, we introduce a novel extension to pink noise, in the form of Spatio-Temporal Noise. This variation enhances traditional temporally correlated noise by dynamically scaling its magnitude based on the exploration status of the current state.

In summary, this paper presents the following findings:

- Pink Noise is indeed a better default than both traditionally used noise types, i.e. White Noise and OU Noise, as is claimed by Eberhard et al. (2023). We report our findings here.

- Noise schedulers are found to be comparable to Pink Noise in a majority of the environments, and out-perform Pink Noise in certain environments, contrary to the Eberhard et al. (2023)'s findings. We also offer added baselines in the form of comparisons between different types of schedulers.

- Spatio-Temporal Noise enhances the results given by standard Pink Noise and we present it as another possible alternative for a default noise. We also report comparisons between all of the above.

## 2 Background and Related Work

### 2.1 Reinforcement Learning

We establish a discounted Markov Decision Process (MDP) formulation of RL, defined by the tuple $(\mathcal{S}, \mathcal{A}, P, \mathcal{R}, \gamma)$, where $\mathcal{S}$ is the state space; $\mathcal{A}$ is the action space; $\mathcal{R}$ is the set of possible rewards; $\gamma \in [0,1]$ is the discount factor; and $P : S \times \mathcal{R} \times \mathcal{S} \times \mathcal{A} \to [0,1]$ is the state transition probability function. For a given state $s \in \mathcal{S}$ and action $a \in \mathcal{A}$, the agent transitions to state $s'$ gaining a reward $r \in \mathcal{R}$ with the probability $P(s', r|s, a)$.

The action-value function $q_\pi$ of control policy $\pi$ is defined as:

$$q_\pi(s, a) = \mathbb{E}_\pi \left[ \sum_{k=0}^\infty \gamma^k r_{t+k+1} | S_t = s, A_t = a \right]$$

The goal of the agent is to learn an optimal control policy $\pi$ that maximizes the expected reward over time.

### 2.2 Exploration

In order to learn the most optimal policy, the agent needs to explore the action space while maximizing the rewards it gets. The exploration-exploitation tradeoff is a fundamental notion in Reinforcement Learning, where the agent has to choose between the greediest policy that has already been explored and unexplored policies that could potentially give higher rewards. In continuous action spaces, we aim to map the start state with the best behavior or action by implementing a policy search.

On-policy methods, such as TRPO (Schulman et al., 2015) and PPO (Schulman et al., 2017) utilize the results from previous iterations to improve the next iteration. They evaluate and improve the same policy that is used for exploration. Off-policy methods, such as TD3 (Fujimoto et al., 2018), SAC (Haarnoja et al., 2018) and MPO (Abdolmaleki et al., 2018), learn from policies that may be different from the policy used for action selection. This allows them to explore while learning the optimal policy simultaneously.

Q-learning off-policy algorithms like DDPG (Lillicrap et al., 2019), and TD3 learn a Q-function and a policy concurrently, where these algorithms are specifically adapted for continuous action spaces. Algorithms like SAC and MPO differ from the former two as they train stochastic policies, whereas DDPG and TD3 train deterministic policies.

SAC is an actor-critic learning framework that aims to maximize rewards while maximizing entropy. Actor-critic algorithms alternate between policy evaluation and policy improvement based on a value function. MPO integrates Expectation Maximization (EM) techniques to optimize the policy without relying on the gradient of the Q-function for updates. TD3 is an extension of DDPG, which improves upon it by learning two Q-value functions and uses the minimum value function estimate during policy updates. We utilize mainly these algorithms to benchmark our methods.

### 2.3 Noise and Randomness

Adding randomness to a policy proves to be the easiest path to improving exploration. A common strategy in Q-learning algorithms in discrete control is the epsilon greedy strategy, where a random action is chosen with a certain probability $\epsilon$ that is usually decayed with time. In the continuous domain, exploration is accomplished by various strategies, including rewarding exploration through novelty bonuses (Tang et al., 2017), bandit method-inspired optimistic action selection based on Q-function uncertainty (Osband et al., 2016), and directly integrating randomness into policy parameters ((Mania et al., 2018), (Plappert et al., 2017)) or actions.

The addition of noise can introduce randomness. It can be independently sampled from a stochastic distribution or directly integrated into the policy. Noise can be temporally correlated or state-dependent (Rückstieß et al., 2008), promoting behavior that leads to more exploration in lesser explored spaces. This noise is commonly injected into the action space, but it can also be added directly into the agent's parameters (Plappert et al., 2017).

### 2.4 Colored Noise

In deterministic off-policy algorithms like DDPG and TD3, action noise is added to the policy as:

$$a_t = \mu(s_t) + \sigma\epsilon_t$$

where $\mu$ is the policy, $\sigma$ is the scale parameter, and $\epsilon_{1:T} = (\epsilon_1, ..., \epsilon_T)$ represents the *noise* sampled randomly from distributions specific to the type of noise. In the case of Gaussian noise or **White noise (WN)**, $\epsilon_t$ is sampled independently at every time step from a normal distribution with $\mu = 0$, $\sigma = 0.1$, and $\Sigma = \mathbf{I} \cdot \sigma$.

Stochastic policy algorithms like SAC and MPO also utilize action noise. In continuous action spaces, these algorithms commonly adopt a diagonal Gaussian policy distribution, which is represented as:

$$a_t \sim \mathcal{N}\left(\mu\left(s_t\right), \mathrm{diag}\left(\sigma\left(s_t\right)\right)^2\right)$$
$$a_t = \mu\left(s_t\right) + \sigma\left(s_t\right) \odot \varepsilon_t$$

where $\epsilon_t \sim \mathcal{N}(0, I)$. Again, $\epsilon_{1:T}$ is Gaussian white noise, scaled by the function $\sigma$.

Another popular choice of noise is **Ornstein-Uhlenbeck (OU) noise** (Uhlenbeck & Ornstein, 1930), wherein $\epsilon_t$ is sampled from the following temporal process:

$$\varepsilon_{a_t} = \varepsilon_{a_{t-1}} + \theta\left(\mu - \varepsilon_{a_{t-1}}\right) \cdot \mathrm{d}t + \sigma\sqrt{\mathrm{d}t} \cdot \epsilon_t$$
$$\varepsilon_{a_0} = 0 \quad \epsilon_t \sim \mathcal{N}(0, I)$$

OU noise is usually generated with $\theta = 0.15$ and is very close to integrated white noise known as Brownian motion. Eberhard et al. (2023) define a family of temporally correlated noises called *colored noise*, characterized by a parameter $\beta$, which determines the degree of temporal correlation. In this context, white noise has $\beta = 0$, Brownian motion (red noise) has $\beta = 2$, and **pink noise**, which is halfway between the two, has $\beta = 1$.

## 3 Experiments

We conduct three distinct sets of experiments, to verify and extend the results of the Eberhard et al. (2023). All experiments were run on continuous control environments taken from DeepMind Control Suite Tassa et al. (2018) and OpenAI Gym Brockman et al. (2016) (listed in Table 3), as in the original paper. We evaluate algorithms and noise types on two metrics, Mean Performance and Final Performance, which are the returns averaged over all training steps and the last five percent of the training steps respectively (For details, refer A.1).

- First, we reproduce the original paper's results by averaging over 5 seeds, utilizing the SAC and MPO algorithms. This set of experiments attempts to reproduce the original paper's returns.

- Second, we evaluate three noise schedulers, atanh, cosine, and linear, against pink noise, on the SAC algorithm. Here, our main goal is to understand where pink noise lies in comparison with schedulers.

- Third, we propose an alternative noise addition algorithm - spatio-temporal noise - and compare it with pink noise for the SAC algorithm.

### 3.1 Pink Noise as a Default

We first attempt to reproduce the results shown by the original paper by investigating the effect that using pink noise has on the SAC and MPO algorithms. We conclude that on average, pink noise gives better results than just using white noise or OU noise, and is, therefore, the more suitable default noise type. Our detailed results are tabulated along with the authors' results in 1. The training plots are plotted in 6 and 7.

### 3.2 Pink Noise for Exploration

To gain insight into why pink noise is a good default noise distribution for exploring environments with unknown dynamics, we experiment with an agent in bounded integrator and oscillator environments, where its actions are driven solely by pink noise. We expect this experiment to reveal how the specific frequency components of pink noise help the agent effectively explore these diverse environments. While pink noise may not maximize exploration in every scenario, Fig 1 demonstrates that it consistently outperforms or performs competitively with white and OU noise across diverse tasks. This suggests its suitability as a general-purpose exploration strategy while acknowledging that alternative noise distributions might be preferable in specific cases. See Sec A.3 for more detailed results.

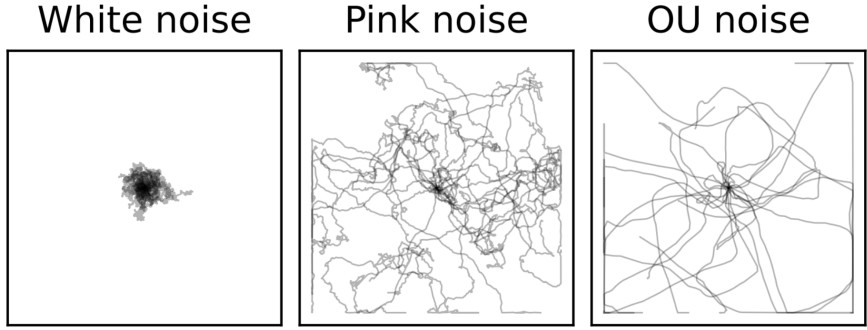

Figure 1: Exploration done by White, Pink, and OU Noise types in the Bounded Integrator environment, where exploration is purely controlled by noise and the environment is constrained by a square boundary. It is clear that the most optimal exploration is achieved when pink noise is used.

### 3.3 Color Scheduler vs Pink Noise

The variation in returns across different noise distributions stems from their temporal correlation characteristics. It is also dependent on the environment's specifications and its preference for temporal correlation. One method of bringing about this difference is to vary the value of $\beta$ across timesteps. These techniques are called Noise Schedulers, and they can be of multiple kinds depending on the function that $\beta$ is decayed with. We experiment with linear, atanh, and cosine schedulers, varying $\beta$ from 2 to 0[1].

The authors used a linear scheduler that cycles between white noise and red noise. Instead, we use a scheduler that shifts from red to white noise over the total time steps, since changing beta values rapidly could cause problems concerning noise generation.

We find that the noise schedulers (A.2) match pink noise in almost all environments (see Table 2). However, caution is to be exercised while using extremely fast-decaying schedulers such as atanh. In environments

---

[1]To counter the problem of atanh not resulting in non-zero returns in multiple environments due to decaying *too* fast, it is instead decayed from 2.5 to 0.

that require high levels of global exploration, fast-decaying schedulers can lead to insufficient exploration. This could result in the model losing out on further off-reward states, that it would have otherwise explored while using pink or OU noise.

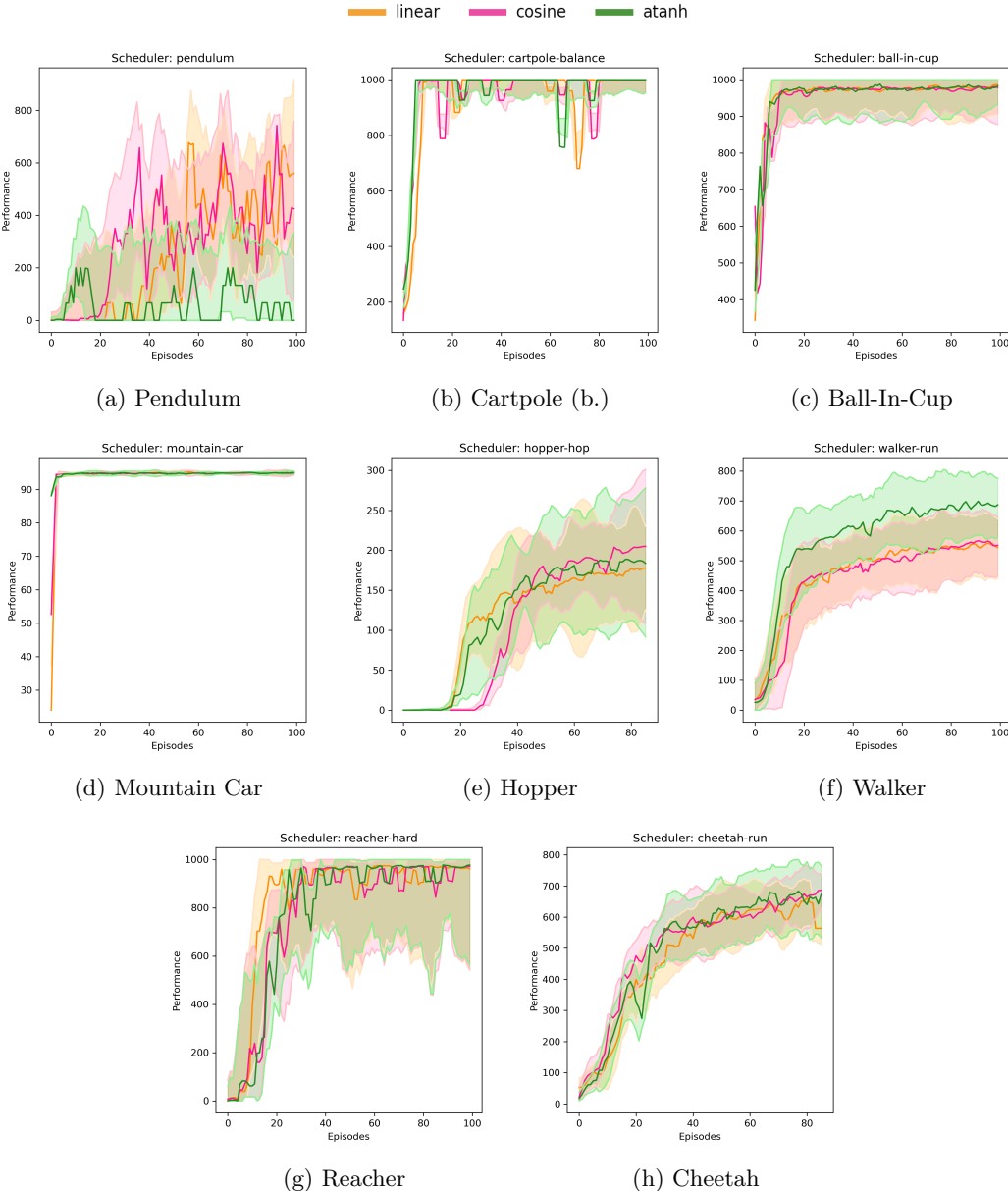

Figure 2: Performance comparison of atanh, cosine, and linear schedulers (A.2), on 8 different environments, utilizing the SAC algorithm. Although atanh performs better in the Walker environment(f), it underperforms by a great margin in the Pendulum environment(a) due to its fast-decaying nature.

Table 1: Our performance comparison of SAC and MPO on all environments using White Noise, OU Noise, and Pink Noise, using two metrics, final and mean performance, averaged over all seeds. Along with a comparison with the theoretical best noise type for an environment - Oracle - and theoretical worst - Anti, and Gain is the difference achievable by changing noise type. We observe that pink noise on average performs at least as good, if not better, than other noise types. See Fig 6 and Fig 7 for plots.

| Environment | Agent | Final Performance | | | Mean Performance | | | | | |
|---|---|---|---|---|---|---|---|---|---|---|
| | | WN | OU | Pink | WN | OU | Pink | Oracle | Anti | Gain |
| Pendulum | MPO | 21 | 532 | 387 | 38 | 405 | 297 | 670 | 239 | 430 |
| | SAC | 561 | 695 | 469 | 168 | 589 | 422 | 361 | 158 | 202 |
| | MPO$^2$ | 311 | 702 | 574 | 247 | 651 | 558 | 670 | 239 | 430 |
| | SAC$^2$ | 224 | 350 | 446 | 158 | 283 | 294 | 361 | 158 | 202 |
| Cartpole (b.) | MPO | 989 | 987 | 939 | 917 | 858 | 886 | 967 | 928 | 39 |
| | SAC | 1000 | 967 | 906 | 964 | 940 | 944 | 950 | 890 | 59 |
| | MPO$^2$ | 999 | 1000 | 1000 | 928 | 940 | 967 | 967 | 928 | 39 |
| | SAC$^2$ | 960 | 908 | 958 | 939 | 890 | 941 | 950 | 890 | 59 |
| Cartpole (s.) | MPO | 769 | 359 | 238 | 621 | 194 | 176 | 666 | 489 | 177 |
| | SAC | 217 | 562 | 636 | 209 | 527 | 566 | 533 | 159 | 374 |
| | MPO$^2$ | 703 | 784 | 788 | 535 | 499 | 666 | 666 | 489 | 177 |
| | SAC$^2$ | 377 | 608 | 730 | 226 | 459 | 532 | 533 | 159 | 374 |
| Ball-In-Cup | MPO | 957 | 969 | 961 | 902 | 868 | 914 | 948 | 909 | 39 |
| | SAC | 976 | 971 | 971 | 958 | 925 | 947 | 941 | 901 | 39 |
| | MPO$^2$ | 974 | 973 | 978 | 926 | 909 | 948 | 948 | 909 | 39 |
| | SAC$^2$ | 976 | 975 | 979 | 930 | 901 | 933 | 941 | 901 | 39 |
| MountainCar | MPO | 0 | 93 | 93 | 0 | 92 | 92 | 92 | 13 | 78 |
| | SAC | 0 | 94 | 94 | 0 | 94 | 94 | 93 | 0 | 93 |
| | MPO$^2$ | 13 | 56 | 92 | 13 | 52 | 91 | 92 | 13 | 78 |
| | SAC$^2$ | 0 | 90 | 94 | 0 | 89 | 93 | 93 | 0 | 93 |
| Hopper | MPO | 1 | 26 | 30 | 1 | 13 | 23 | 69 | 14 | 54 |
| | SAC | 33 | 146 | 128 | 27 | 88 | 88 | 80 | 43 | 36 |
| | MPO$^2$ | 25 | 62 | 108 | 14 | 34 | 69 | 69 | 14 | 54 |
| | SAC$^2$ | 89 | 94 | 119 | 43 | 53 | 77 | 80 | 43 | 36 |
| Walker | MPO | 925 | 450 | 654 | 782 | 340 | 502 | 390 | 284 | 106 |
| | SAC | 441 | 434 | 549 | 312 | 320 | 454 | 472 | 363 | 108 |
| | MPO$^2$ | 530 | 377 | 448 | 384 | 284 | 363 | 390 | 284 | 106 |
| | SAC$^2$ | 593 | 506 | 602 | 437 | 363 | 471 | 472 | 363 | 108 |
| Reacher | MPO | 952 | 410 | 746 | 867 | 167 | 671 | 888 | 581 | 306 |
| | SAC | 969 | 897 | 971 | 816 | 701 | 781 | 776 | 653 | 122 |
| | MPO$^2$ | 956 | 856 | 966 | 864 | 600 | 871 | 888 | 581 | 306 |
| | SAC$^2$ | 955 | 914 | 940 | 776 | 653 | 745 | 776 | 653 | 122 |
| Cheetah | MPO | 577 | 241 | 419 | 417 | 229 | 318 | 543 | 440 | 103 |
| | SAC | 578 | 553 | 654 | 446 | 435 | 472 | 502 | 439 | 63 |
| | MPO$^2$ | 666 | 612 | 678 | 481 | 440 | 543 | 543 | 440 | 103 |
| | SAC$^2$ | 631 | 577 | 640 | 469 | 439 | 483 | 502 | 439 | 63 |

---

[2]These are the Eberhard et al. (2023)' results.

### 3.4 Spatio-Temporal Noise

Our paper also experiments with a combination of spatial and temporal conditioning to the noise. We do this by adding a scaling factor, as shown in Algorithm 1, to the temporally dependent noise. The scaling factor is initialized to a value $h$, which is then decayed to 1 based on the state's visitation. We model the decay using a reverse sigmoid function and then fix the decay timestep as a hyperparameter $t$. We use a binning algorithm based on k-nearest neighbors that store a mean and an associated frequency. At every stage, we then use the bins to estimate similar states visited, which we pass as the decay step to determine the scaling factor. In our experiments, we find that this setup converges to returns better than pink noise in 7 of 8 environments while being on par in the last one (see Table 2 and Fig 3).

---

**Algorithm 1:** Spatio-Sampling algorithm which returns the number of closest neighbors, which is passed to the decay function to determine the scaling factor $h$ for the temporal noise. $B_{\max}$ and $r_{\min}$ are hyperparameters that represent the maximum number of bins to maintain and the minimum radius to determine the proximity of neighbors respectively.

---

**Input:** Bins & Weights $\langle B, W \rangle = [(B_1, W_1), (B_2, W_2), \ldots, (B_k, W_k)]$, Current State $S$

Initialize $N \leftarrow 0$, $r_{\min} \in \mathbb{R}$, $d_{\text{least}} \leftarrow \infty$

// If number of bins is less than $B_{\text{max}}$ then make a new bin

**if** $K < B_{\max}$ **then**
    $\langle B, W \rangle \leftarrow \langle B, W \rangle + (S, 1)$
    return 1

// Iterate through bins to find the closest bin to given
// state $S$ and number of bins within $r_{\text{min}}$ distance

**for** $(B_i, W_i) \in \langle B, W \rangle$ **do**
    $d \leftarrow |B_i - S|$
    $d_{\text{least}} \leftarrow \min(d_{\text{least}}, d)$
    **if** $d < r_{\min}$ **then**
        $N \leftarrow N + W_i$

// $N$ now stores the number of nearest neighbors
// We also accordingly add the new state to its closest bin

$i \leftarrow$ index of closest bin

$B_i \leftarrow \frac{B_i \cdot W_i + S}{W_i + 1}$

$W_i \leftarrow W_i + 1$

return $N$

**end**

---

Table 2: Final performance of various noise addition techniques - pink noise, schedulers, and spatio-temporal noise - using SAC algorithm. From these results, we conclude that spatio-temporal noise is an improvement over standard pink noise.

| Environment | Final Performance | | | | |
|---|---|---|---|---|---|
| | **Pink** | **Atanh** | **Cosine** | **Linear** | **Spatio-Temporal** |
| Ball-In-Cup | 971 | 978 | 978 | 982 | 982 |
| Cartpole (b.) | 906 | 1000 | 1000 | 1000 | 1000 |
| Pendulum | 469 | 40 | 406 | 585 | 507 |
| Reacher | 971 | 970 | 969 | 963 | 963 |
| Walker | 549 | 688 | 555 | 551 | 525 |
| Cheetah | 654 | 705 | 727 | 642 | 742 |
| Hopper | 128 | 191 | 193 | 182 | 134 |
| MountainCar | 94 | 95 | 94 | 94 | 95 |

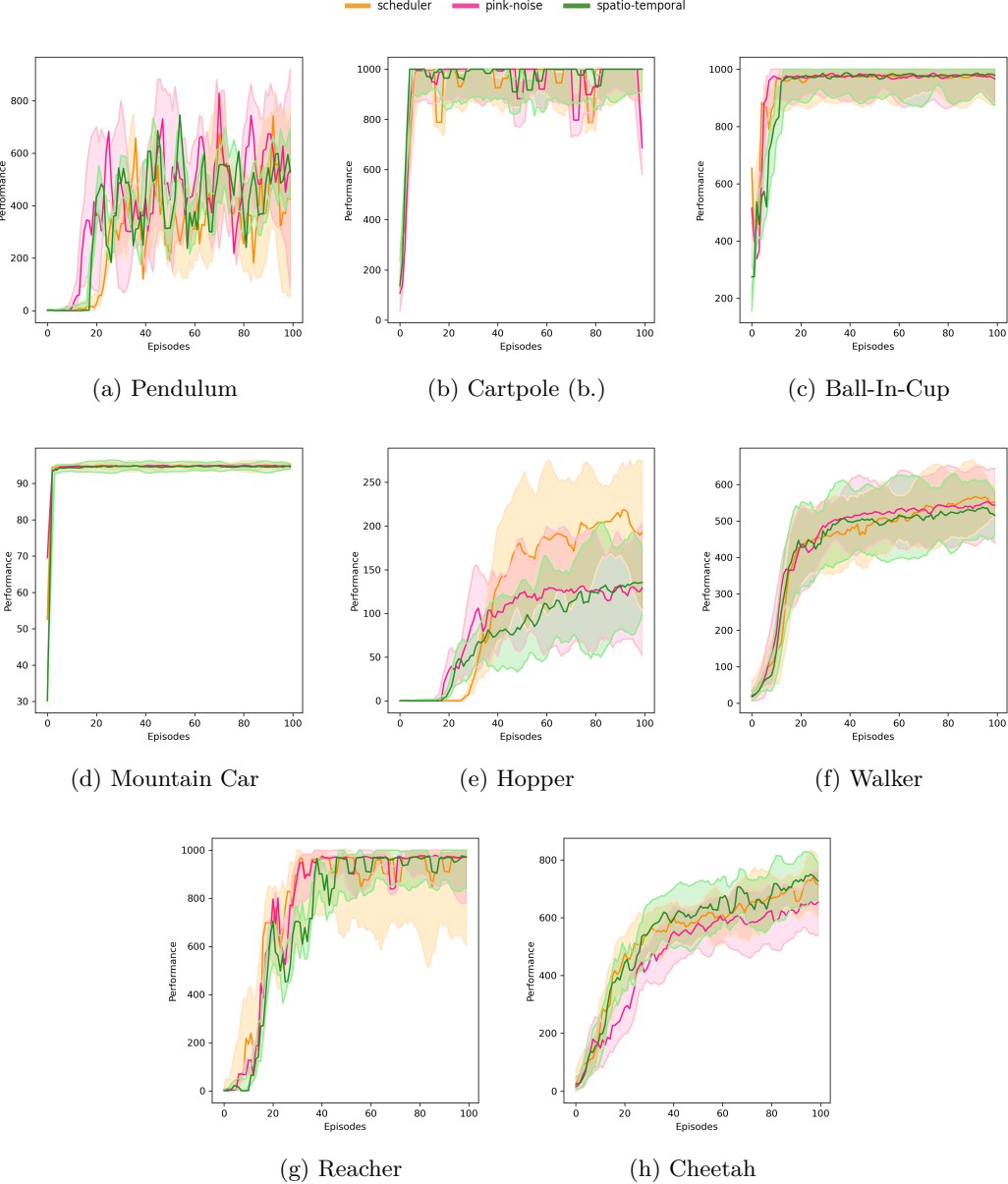

Figure 3: Performance comparison of spatio-temporal noise, pink noise, and cosine noise-scheduler[3]on 8 different environments. We note that spatio-temporal noise is outperformed by the cosine noise scheduler in environments like Hopper. Its final performance is just as good, if not better than pink in all 8 environments.

## 4  Discussion

Local exploration involves making short-term decisions to optimize rewards, focusing on immediate consequences, while global exploration, or deep exploration, considers the future implications of decisions. Global exploration addresses the challenge of maintaining motivation and optimizing decision-making strategies over extended periods. This approach also acknowledges that overall rewards might diminish rapidly within environments with sparse rewards, leaving the agent without sustained long-term incentives.

Employing uncorrelated noise, such as white noise, facilitates local exploration, whereas the application of highly correlated noise, such as OU noise, enables adequate exploration necessary for overcoming potential entrapment in local optima. However, the exclusive use of highly correlated OU noise introduces an alterna-

tive challenge, characterized by the generation of trajectories that are strongly off-policy. Strongly correlated action noise, such as red noise, can help with under-actuated environments where we need more global exploration. Environments that neither have a vast state space nor exhibit under-actuation do not profit from highly correlated noise. Instead, the temporally correlated noise might lead to off-policy trajectory data, inhibiting learning. Therefore, neither OU noise nor white noise acts as reasonable defaults.

The power spectral density of red noise is lower at higher frequencies. This means that low-frequency components dominate the signal. Since the noise signal lacks high frequency components, it does not fluctuate a lot leading to a much smoother signal. This is why red noise is strongly time-correlated. Pink noise also has a similar power spectral density, just that the difference in magnitudes at high frequency and low frequencies are not as high as in red noise, leading to a weaker time correlation. Thus, pink noise can be seen as the middle ground between white and OU noise that works well for all environments, therefore acting as a good default.

Intuitively, the model should initially explore globally, as it is not familiar with the environment. Later on, when the model gains familiarity with the environment it should tone down its global exploration, and try to explore locally. This is the idea behind color schedulers. Eberhard et al. (2023) experiment with a linear noise scheduler, starting with just "red" noise and moving linearly towards white noise as we explore further.

Adjusting the amount of exploration done by the agent based on the number of similar states that we have already visited could yield even more efficient exploration. This led us to experiment with a combination of spatial and temporal conditioning to the noise. We do this by adding a scaling factor to the temporally dependent noise, that is varied according to the number of similar states. We used 5 random seeds and observed a large amount of variance in the MPO results. This leads us to believe that the agent is not able to find a close-to-optimal policy every single time due to it not being able to explore thoroughly while using pink noise with the MPO algorithm.

## 5 Conclusion

In conclusion, this paper explores the role of colored noise, specifically pink noise, as a default exploration strategy for off-policy reinforcement learning agents navigating continuous control spaces. Through a series of experiments on diverse environments, we find that pink noise generally outperforms traditional white and Ornstein-Uhlenbeck (OU) noise, validating the claims made by Eberhard et al. (2023). The intermediate nature of pink noise seems to strike a balance between local and global exploration, making it a promising default choice.

Moreover, we investigate the effectiveness of noise schedulers, namely, linear, atanh, and cosine, and find that they match pink noise's returns in every environment that we experiment on while outperforming it in the hopper environment. Contrary to the original authors' conclusion we also put forward the noise scheduler as a good default choice as an action noise distribution. The type of scheduler used usually does not make a drastic difference in the performance but the returns that we see for atanh on the pendulum environment lead us to choose linear and cosine as the better options.

Additionally, we propose a novel exploration technique, spatio-temporal noise, which combines both spatial and temporal conditioning. This approach, incorporating a scaling factor based on the state's visitation frequency, outperforms traditional pink noise in almost all tested environments. The results highlight the potential of spatio-temporal noise in achieving efficient exploration and suggest its consideration as a default exploration strategy.

In summary, this paper contributes valuable insights into exploration strategies for reinforcement learning agents, showcasing the potential benefits of incorporating spatial and temporal information into the exploration process. We put forward three noise distributions namely pink noise, spatio-temporal noise, and noise scheduler as candidates for default action noise distributions. These would be good choices when we do not know the characteristics of the environment and do not want the noise distribution to have a negative effect on the returns. We can at least conclude that for most environments the choice of action noise among the three named above should not have a huge effect on the returns. Our findings open avenues for further research in tailoring exploration strategies based on the specific characteristics of diverse environments.

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

# A  Appendix

## A.1  Experiment details

All experiments were conducted on SAC and MPO on all environments listed in 3. The training consists of 100 episodes of 10,000 steps each, where the reward for each episode is averaged over five evaluation rollouts. The mean performance is calculated by averaging rewards over all the episodes, whereas the final performance is calculated by averaging the rewards obtained in the last five percent of the runs. Our novel and scheduler experiments were conducted on all environments other than Cartpole-Swingup due to computing constraints. We use the following nine environments for reproducibility.

Table 3: Environments used

| Environment | Source | ID |
|---|---|:---:|
| Pendulum | DMC | pendulum (swingup) |
| Cartpole (b.) | DMC | cartpole (balance_sparse) |
| Cartpole (s.) | DMC | cartpole (swingup_sparse) |
| Ball-In-Cup | DMC | ball_in_cup (catch) |
| MountainCar | Gym | MountainCarContinuous-v0 |
| Hopper | DMC | hopper (hop) |
| Walker | DMC | walker (run) |
| Reacher | DMC | reacher (hard) |
| Cheetah | DMC | cheetah (run) |

## A.2  Noise Schedulers

Here we provide details of each of our schedulers

### A.2.1  Linear Scheduler

The linear noise scheduler decays $\beta$ from 2 to 0 linearly. This is modeled by the function

$$\beta = (1 - x) * 2$$

where x represents the proportion of training completed.

### A.2.2  Cosine scheduler

The Cosine noise Scheduler decays $\beta$ from 2 to 0 following a cosine curve. This is modeled by the function

$$\beta = 1 + \cos(\pi \cdot x)$$

where x represents the proportion of training completed.

### A.2.3  atanh schedulers

The atanh noise scheduler decays $\beta$ from 2.647(approx) to 0 following a arctanh curve. This is modeled by the function

$$\beta = \operatorname{arctanh}(\min(1 - x, 1 - 10^{-6}))$$

where x represents the proportion of training completed.

### A.3 Bounded Integrator and Oscillator Environments

The bounded integrator environment is a particle bounded in a square that it is unable to leave and is controlled by giving it a velocity.

$$s_{t+1} = \text{clip}(s_t + a_t, -c, +c)$$

where $s$, the state is position, $a$, the action is velocity, and $c$ is the side of the square the particle is bounded in. The policy is executed, whereby the velocities along distinct axes are independently controlled, and the action is just noise with unit variance. This is why it is a good representation of the exploration done by the different types of action noise. The code for the bounded integrator environment was taken from the supplementary material provided by Eberhard et al. (2023).

The oscillator environment is a simple damped harmonic oscillator, consisting of a particle of mass $m$ connected to a spring of stiffness $k$, acted upon by a force $F$, where the friction coefficient is set to 0, resulting in the differential equation

$$mx^{''} = F - kx$$

where $x^{''}$ represents the acceleration and $x$ represents the position. The system's resonant frequency, $f$, which is set by adjusting the stiffness and mass, defines the oscillator environment.

The state space consists of the particle's position and velocity, and the action is the force driving it. The images display the sensitivity of the types of action noise to the parameterization of time-driven elements in the environment.

We have used the gym environment for the oscillator environment given by Eberhard et al. (2023) at https://github.com/onnoeberhard/oscillator-gym

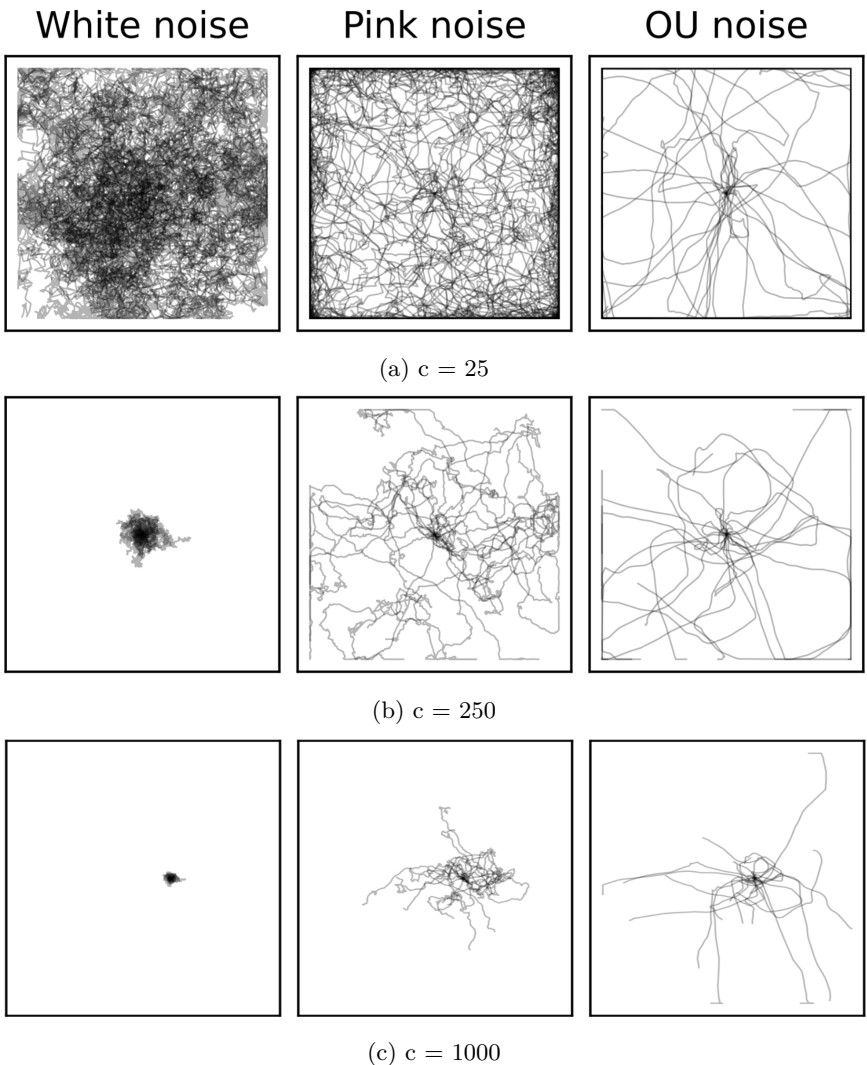

Figure 4: Results of white, pink, and OU noise on the Bounded Integrator environment, where $c$ represents the side of the bounding square. It is observable that pink noise generates the most balanced exploration patterns.

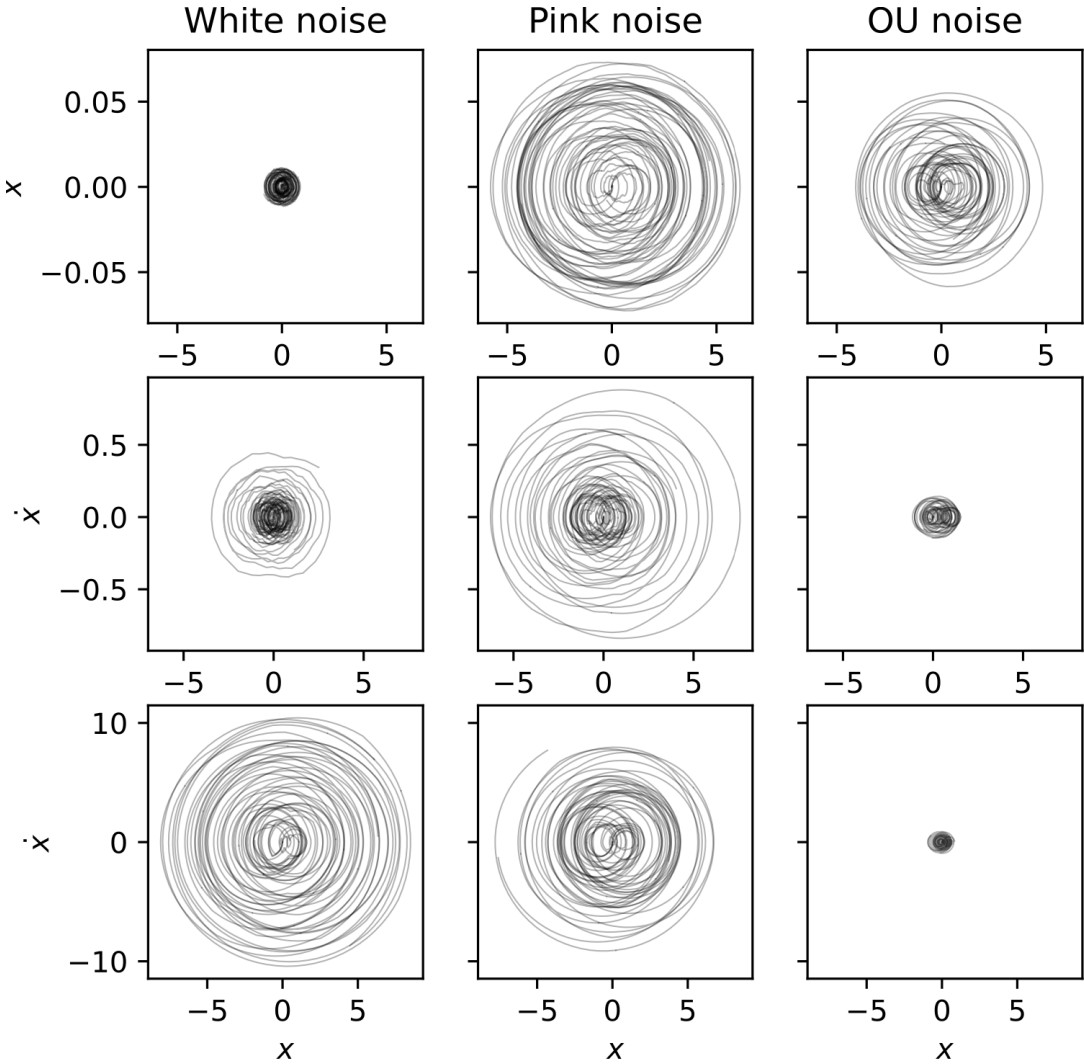

Figure 5: Results of white, pink, and OU noise on the Oscillator Environment for $f \in \{0.2, 0.02, 0.002\}$. It is visible that pink noise is the least afflicted by the variation in resonant frequency $f$ of the harmonic oscillator.

## A.4 Performance of Pink Noise using MPO

Here are the detailed plots for the results of utilizing the MPO algorithm with white, pink, and OU noise in 9 different environments. The implementation was achieved by using the Tonic RL library (Pardo (2020)).

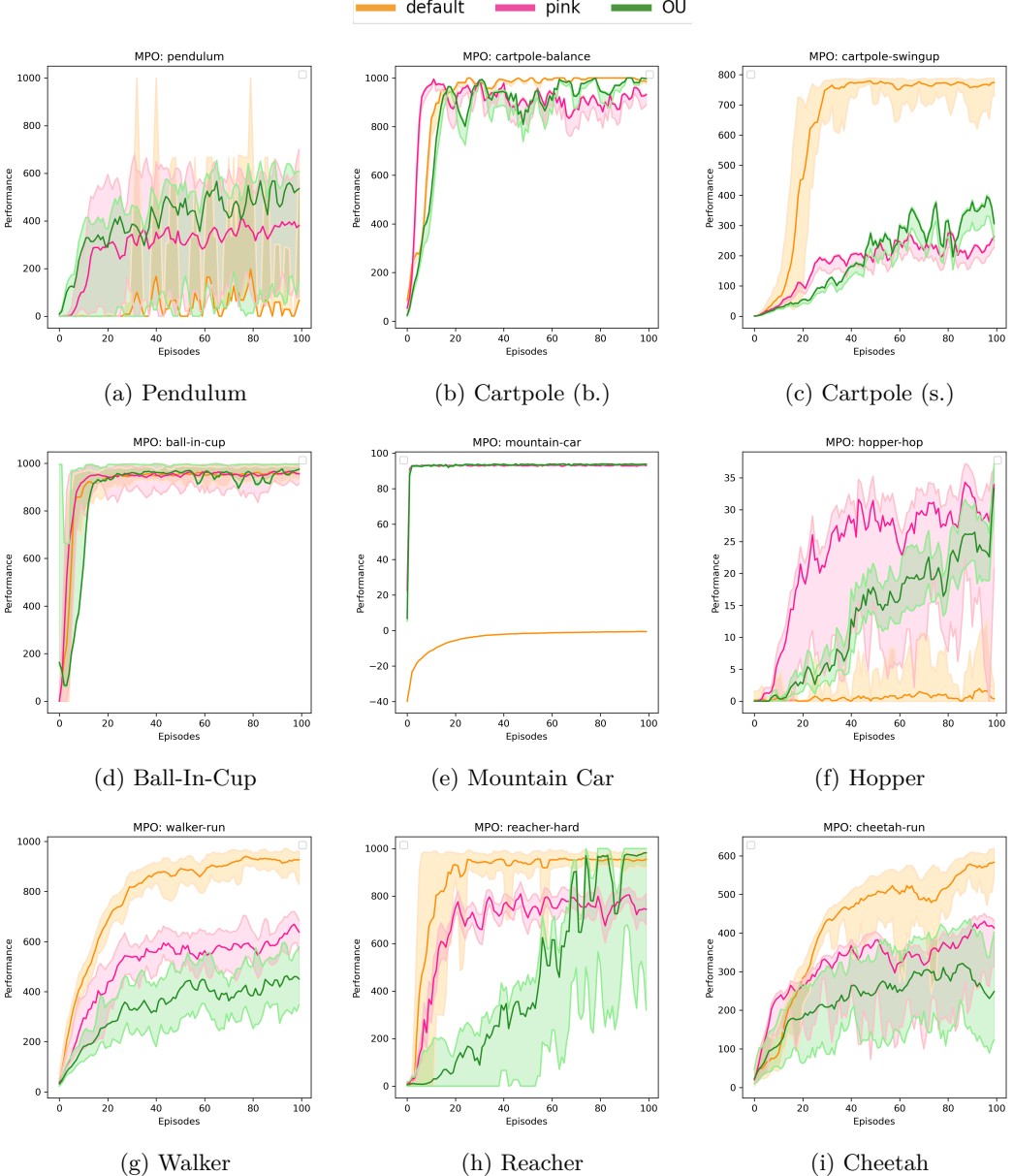

Figure 6: Performance comparison of white, pink, and OU noise, utilizing the MPO algorithm on 9 different environments, averaged across 5 seeds. We conclude that pink noise is the best default among the three.

## A.5 Performance of Pink Noise using SAC

Here are the detailed plots for the results of utilizing the SAC algorithm with white, pink, and OU noise in 9 different environments. The implementation was achieved by using the stable_baselines3 library (Raffin et al. (2021)).

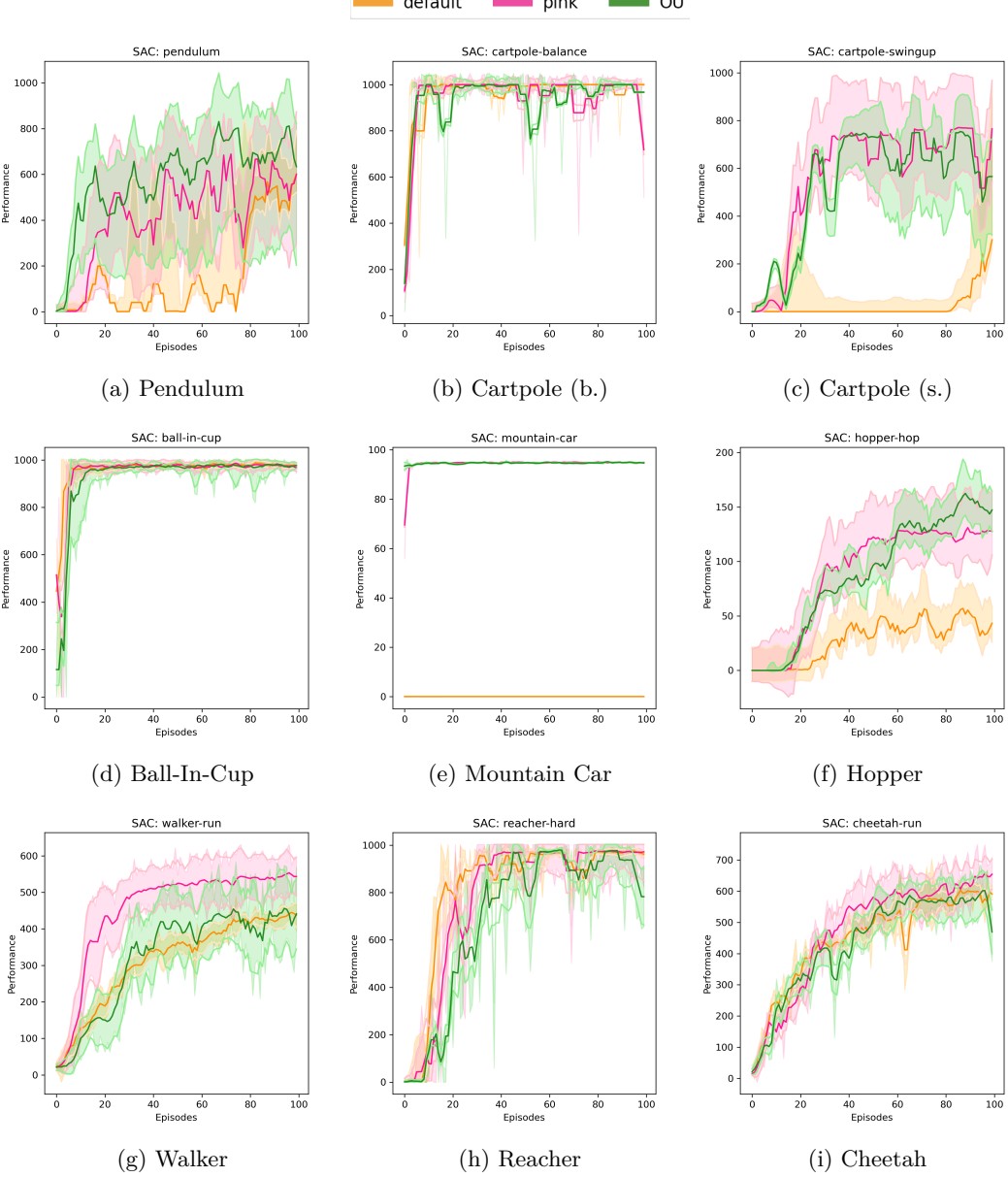

Figure 7: Performance comparison of white, pink, and OU noise, utilizing the SAC algorithm on 9 different environments, averaged across 5 seeds. We conclude that pink noise is the best default among the three.

