# OpenReview forum: "Exploring Exploration: A Comparative Analysis of Colored Noise Strategies in Reinforcement Learning"
_TMLR — Rejected by TMLR_

### Review · Reviewer_wSqx · 2024-04-03

**Summary Of Contributions:**

# Summary
The paper _Exploring Exploration: A Comparative Analysis of Colored Noise Strategies in Reinforcement Learning_ presents an empirical study of coloured noise strategies -- a family of functions which produce temporally correlated noise [1] -- in off-policy actor-critic. The paper specifically considers white noise, OU (red) noise, and pink noise, an interpolation between white and red noise. The paper further studies a number of pink noise schedules and how such schedules can influence the performance of RL algorithms. Finally, the paper presents a novel temporal-spatial noise scheduling algorithm which introduces noise correlated both temporally and spatially.


# References

[1] Onno Eberhard, Jakob Hollenstein, Cristina Pinneri, Georg Martius. Pink Noise Is All You Need: Colored Noise Exploration in Deep Reinforcement Learning. International Conference on Learning Representations, 2023.

**Audience:**

Yes

**Broader Impact Concerns:**

No broader impact concerns.

**Claims And Evidence:**

No

**Requested Changes:**

In order to secure my recommendation for acceptance, the paper would have to conduct a rigorous empirical study as outlined in my **Main Argument** above. To reiterate, the paper should utilize many more random seeds with estimates of confidence for which assumptions are satisfied.

**Strengths And Weaknesses:**

# Main Argument

**Recommendation: Reject**

My overall concern with this paper is the empirical analysis. This paper presents an empirical study of action noise in off-policy actor-critic, but the claims are not supported by solid empirical evidence. Because the paper utilizes too few seeds, the shaded regions in nearly all plots are overlapping significantly, indicating little to no statistical significance in results. Yet, strong claims are made. For example, in section 3.1 the following claim is made

> We conclude that on average pink noise gives better results than just using white noise or OU noise, and is, therefore, the more suitable default noise type.

in reference to figures 6 and 7. Yet the shaded regions plotted in these figures overlap for nearly all noise types across environments, indicating a lack of statistical significance in performance across noise types. Further, what does it mean to _give better results_ (see below)?

Further, I could not find any indication in the paper of what the shaded regions denoted in the figures. I therefore assume that these shaded regions denote standard errors. If this is the case, then confidence intervals with 95-99% levels of confidence would be nearly double the width of the shaded regions, exacerbating the lack of statistical significance. Further, confidence intervals based on standard errors assume normality, and it is unlikely that the data is normally distributed. Instead of using confidence intervals for which assumptions are not satisfied, my recommendation would be to use **many** more random seeds to construct estimates of confidence for which assumptions are satisfied.

We see similar issues with the experiments in section 3.3 (noise scheduling) and section 3.4 (spatio-temporal noise) -- nearly all shaded regions are overlapping with strong conclusions drawn. For these experiments, I could not find indication of how many runs were used, and so I assume that 5 runs were also used as done in the previous experiment. For example, in section 3.3, the claim is made that _(a)_ the $atanh$ noise schedule performs best in the Walker environment and _(b)_ that the "cosine [noise schedule] is the most consistent among all schedulers". Yet in both cases, shaded regions are significantly overlapping. In regards to section 3.3, the paper mentions that their conclusions are contrary to the findings of [1]. I would expect this is due to the use of too few seeds and improper estimates of confidence. In section 3.4, the claim is made that the spatio-temporal noise converges to higher returns than pink noise, but the presented results have nearly all shaded regions overlapping.  Estimates of confidence are not reported in Table 1 or Table 2 and I suspect that nearly all confidence intervals would be overlapping here.

What I have mentioned in the preceding paragraphs is only a subset of the problematic claims made in the paper. As I have already stated, my recommendation would be to use many more random seeds with estimates of confidence for which assumptions are satisfied. As already discussed, this is my main concern with the paper. Below, I mention a few secondary concerns.

## Secondary Concerns

The purpose of section 3.1 is not well-substantiated, as reproducing existing results is not in and of itself a novel contribution. What really is the purpose of this experiment? Is it simply to reproduce existing results? The current paper is really an empirical investigation into action noise. Instead of attempting to reproduce current results, it would be better to provide a rigorous empirical evaluation of these different noise types.

Key information for a full understanding of the empirical work is missing. How were hyperparameters chosen? Were they swept? If so, how many seed were they swept for? It is well known that actor-critic algorithms are sensitive to hyperparameters. If hyperparameters were not swept, it could be the case that the ones used were more suitable for one noise type over another. The paper does not outline the equations used for the noise schedulers. How exactly do these schedulers work? Why does the cosine scheduler not result in oscillations?

Overall, the paper is not well-polished:
- The paper often uses the term _original authors_, but this has not been defined. I assume this is in reference to [1], in which case it would be better to use the author's names with a citation.
- What are the bounded integrator and oscillator environments? There was little to no environmental description for either environment.
- In reference to Figure 1, the claim is made that "[pink noise] consistently outperforms or performs competitively with white and OU noise across diverse tasks". How is performance measured here? By visual inspection? Then, in the caption for Figure 1 the claim is made that "It is clear that the most optimal exploration is achieved when pink noise is used". How is exploration quantified? Further, the caption text seems in direct contradiction to the main text.
- The paper often uses the term _performs better_ or _give better results_. What do these terms mean? I suspect these terms mean different things with regards to RL algorithms vs noise types, but the paper never clarifies or defines these terms.
- In reference to figure 4, the paper states that "pink noise generates the most balanced exploration patterns". How is this measured? What does it mean to have a "balanced exploration pattern"? I would argue that in Figure 4(a) white noise exhibits the most effective exploration in terms of state space coverage and placing little density on the state space extremities.
- Sections 3.4 and 5 make contradictory statements. Section 3.4: "we find that this setup [spatio-temporal noise] converges to returns better than pink noise in **7 of 8 environments**...". Section 5: "[spatio-temporal noise] outperforms traditional pink noise in **all tested environments**".
- A claim is made that "**most** off-policy reinforcement learning algorithms utilize the addition of stochastic action noise to the action chosen by the policy". I am not sure this is a true statement.
- Figures 6 and 7 do not have legends

---

> ### Author Response · Authors · 2024-05-07
> **Reply to Reviewer wSqx**
>
> We thank you for your thorough feedback and insightful comments.
>
>  >This paper presents an empirical study of action noise in off-policy actor-critic, but the claims are not supported by solid...
>
> To clarify, we obtain the shaded regions by using the worst value among our runs from each run at each time step as the lower bound and the highest value among our runs at each time step as the upper bound. This, coupled with the fact that adjusting the action noise distribution does not warrant a significant increase in expected reward over pink noise, explains the overlaps between the graphs. This is because pink noise atleast performs decently for every choice of environment that we have previously experimented on.
>
> We do not expect to observe substantial differences in results between the three scheduler types, as they inherently follow a similar approach: they start with an Ornstein-Uhlenbeck (OU) noise-like distribution and gradually shift towards a white noise distribution over time. A small adjustment to the action noise distribution should not significantly impact the returns.
>
> Furthermore, we consider any noise distribution that can at least match the performance of pink noise across all environments and provide higher returns in some environments to be a better result. Our intention is to present pink noise, noise schedulers, and the proposed spatio-temporal noise as viable candidates for action noise distributions, depending on the characteristics of the environment.
>
> We acknowledge that to statistically conclude which noise distribution is a better candidate as a general action noise, considerably more experimentation would be required, which is challenging for us to undertake given our current resources.
>
> >In reference to Figure 1, the claim is made that "[pink noise] consistently outperforms or performs competitively with white...
>
> We explain the statements in our paper accordingly:
>
> Regarding the statement "[pink noise] consistently outperforms or performs competitively with white and OU noise across diverse tasks," we intend to convey that using pink noise does not result in a catastrophically degraded performance in any environment while still providing satisfactory returns in each environment. Examples of catastrophically poor results include the use of white noise in environments such as cartpole and mountain-car.
>
> >In reference to figure 4, the paper states that "pink noise generates the most balanced exploration patterns". How is this measured...
>
> Concerning the statement "It is clear that the most optimal exploration is achieved when pink noise is used," this figure is crucial because it demonstrates the extent of exploration when the action taken by our agent does not change significantly, and the actual change in the action is driven by the noise distribution itself. Here, we aim to show that sampling actions purely from a pink noise distribution leads the agent to cover more space in the environment while satisfactorily exploring areas close to the origin, at intermediate distances from the origin, and far away from the origin. Additionally, this is what we mean by a "balanced exploration pattern."
>
> Regarding the statement "In figure 4, white noise exhibits the most effective exploration in terms of state space coverage and placing little density on the state space extremities," we acknowledge that white noise seems to show better state space coverage only in the first environment with c=25.
>
> >Sections 3.4 and 5 make contradictory statements. Section 3.4: "we find that this setup [spatio-temporal noise] converges...
>
> We have made the correction in our revision, apologies for the oversight.
>
> >A claim is made that "most off-policy reinforcement learning algorithms utilize the addition of stochastic action noise...
>
> As mentioned in works such as "Action Noise in Off-Policy Deep Reinforcement Learning: Impact on Exploration and Performance," additive action noise is commonly used for simple exploration in continuous control domains. For most continuous control setups, action noise is employed to enable simple exploration in off-policy reinforcement learning.
>
> >Figures 6 and 7 do not have legends.
>
> We acknowledge the lack of legends for Figures 6 and 7, which may have hindered the interpretation of the results. In our revised submission, we have included appropriate legends for these figures to enhance clarity and understanding.
>
> We appreciate your detailed feedback, which has highlighted areas where our paper can be improved. We hope we have addressed these concerns and provided a more comprehensive and well-documented presentation of our work in the revised version.
>
> Thank you for your time and valuable input, which has undoubtedly contributed to enhancing the quality and clarity of our paper.

---

### Review · Reviewer_H45F · 2024-04-09

**Summary Of Contributions:**

This paper presents a reproduction of the experiments done in (Eberhard et al., 2023). With the experiments done in DMC and OpenAI Gym continuous control tasks and based on SAC/MPO algorithms, this work demonstrates: (1) Pink noise is a better default exploration noise in comparison with white noise and OU noise, and (2) Noise schedulers are found to be better than Pink noise in most cases.

In addition, the paper propose a new type of noise called Spatio-Temporal Noise by scaling the temporally dependent noise in colored noise, which is empirically evaluated in comparison with pink noise and cosine noise-scheduler.

**Audience:**

Yes

**Claims And Evidence:**

No

**Requested Changes:**

- Provide the detailed equations for the colored noise family in Section 2.4, to make this paper self-contained.
- Provide the exact dynamics of the two environments introduced in Appendix A.2.
- Add more seeds, I think at least 8-10 seeds for each environment. And use *Reliable metrics.*
- Add DMC Quaduped, Humanoid environments, or justify why not using them.
- Add the experiments based on TD3.
- Re-organize the content in Section 3.4. Motivate the proposal method better and provide more details in implementing the methods (e.g., use illustrative plots).
- Justify why Figure 1, 4, 5 look so similar to those in (Eberhard et al., 2023). Is the same code used? In addition, I recommend the authors to use quantitative metrics for these environments. Or at least, provide more discussion on the results to explain in what sense pink noise outperforms the others.
- The legends of Figure 6,7 are missing.
- In Table 1, explain whis the meaning of 'theoretical best/worst' is.

**Strengths And Weaknesses:**

First, the writing is not satisfactory. This paper does not provide enough details and is not self-contained. I think the current organization of content is not proper. The motivation and proposal of the new noise should be presented with an independent section. Currently, it is too short to be clear enough.

Since this paper is a more reproduction report-like paper, the experiments should be convincing enough. However, the experiments have several major issues:

- The choice of environments used are not well explained. Why Quaduped, Humanoid in DMC are not considered? In Appendix A.2, the exact dynamics of the two environments are not provided.
- The comparison results are not convincing. What is the meaning of the error bars of the learning curves reported, e.g., stds? And I found the error bars of the most learning curves are highly overlapped. I do not think a reliable conclusion can be drawn from them. Error bars are not reported in Table 1 as well.
- Figure 1, 4, 5 are highly similar to those in (Eberhard et al., 2023). Besides, I think a quantitative metric should be used to discuss the performance of the different noise types, e.g., comparing state space coverage if the criterion is to explore the space as much as possible.

The proposed method are not motivated sufficiently, and I found it is difficult to understand Algorithm 1 with the text. More explanation should be provided. Finally, similarly the evaluation results in Figure 3 are not convincing enough.

For a reliable empirical evaluation, I recommend the authors to use *Reliable metrics* [1].

---
Reference:

[1] Deep Reinforcement Learning at the Edge of the Statistical Precipice. NeurIPS 2021.

---

> ### Author Response · Authors · 2024-05-07
> **Reply to Reviewer H45F**
>
> We appreciate your thorough feedback and insightful comments. As stated, the primary objective of our work is to reproduce the findings of Eberhard et al. (2023). Additionally, we conducted experiments involving noise schedulers and our proposed spatio-temporal noise approach. The noise schedulers were implemented to span the entire training horizon as a single time period, rather than oscillating over shorter periods. Due to resource constraints, we were unable to extend our experimentation to environments beyond those considered in the original work.
>
> >The choice of environments used are not well explained. Why Quaduped, Humanoid in DMC are not considered? In Appendix A.2, the exact dynamics of the two environments are not provided.
>
> The choice of environments was prioritized to align with Eberhard et al. (2023), as their work served as the foundation for our reproducibility study. Our primary goal was to validate their findings within the same set of environments.
>
> >The comparison results are not convincing. What is the meaning of the error bars of the learning curves reported, e.g., stds? And I found the error bars of the most learning curves are highly overlapped.
>
> We obtain the shaded regions by using the worst value among our runs from each run at each time step as the lower bound and the highest value among our runs at each time step as the upper bound. This, coupled with the fact that adjusting the action noise distribution does not warrant a significant increase in expected reward over pink noise, explains the overlaps between the graphs. This is because pink noise atleast performs decently for every choice of environment that we have previously experimented on.
>
> We do not expect to observe substantial differences in results between the three scheduler types, as they inherently follow a similar approach: they start with an Ornstein-Uhlenbeck (OU) noise-like distribution and gradually shift towards a white noise distribution over time. A small adjustment to the action noise distribution should not significantly impact the returns.
>
> Furthermore, we consider any noise distribution that can at least match the performance of pink noise across all environments and provide higher returns in some environments to be a better result. Our intention is to present pink noise, noise schedulers, and the proposed spatio-temporal noise as viable candidates for action noise distributions, depending on the characteristics of the environment.
>
> We acknowledge that to statistically conclude which noise distribution is a better candidate as a general action noise, considerably more experimentation would be required, which is challenging for us to undertake given our current resources.
>
> >Provide the detailed equations for the colored noise family in Section 2.4, to make this paper self-contained.
>
> We have provided precise equations for the colored noise family in the appendix. Additionally, we have included descriptions of the dynamics for the bounded integrator and oscillator environments in the appendix.
>
> >Justify why Figure 1, 4, 5 look so similar to those in (Eberhard et al., 2023). Is the same code used?
>
> Regarding Figures 1, 4, and 5, we would like to clarify that they were not directly lifted from Eberhard et al. (2023). Instead, these figures were reproduced using the code provided in their supplementary material, applied to different seeds for the bounded integrator environment. We apologize for any confusion, and we have ensured proper attribution and clarification in the revised version.
>
> >The legends of Figure 6,7 are missing.
>
> We acknowledge the lack of legends for Figures 6 and 7, which may have hindered the interpretation of the results. In our revised submission, we have included appropriate legends for these figures to enhance clarity and understanding.
>
> We appreciate your detailed feedback, which has highlighted areas where our paper can be improved. We hope we have addressed these concerns and provided a more comprehensive and well-documented presentation of our work in the revised version.
>
> Thank you for your time and valuable input, which has undoubtedly contributed to enhancing the quality and clarity of our paper.

---

### Review · Reviewer_cEPF · 2024-04-18

**Summary Of Contributions:**

This article tests the effectiveness of several types of policy noise on exploration in continuous control domains. The main noise types evaluated are white noise (completely uncorrelated Gaussian samples), red noise (cumulative noise samples, or Brownian motion), and pink noise (an intermediate between white and red noise), as well as some schedules interpolating between noise types over training time. The noise types are evaluated for two agent algorithms: Soft Actor Critic and MPO. The authors also introduce a new noise type called spatio-temporal noise. The authors show that pink noise performs best for the chosen agents and environments. A schedule of noise performs even better, as well as their proposed spatio-temporal noise. This article follows closely the cited Eberhard 2023 article.

**Audience:**

Yes

**Claims And Evidence:**

No

**Requested Changes:**

Addressing the above issues would improve the article.

**Strengths And Weaknesses:**

# Strengths

The article provides a systematic evaluation of different noise types, agents and environments. The results seem fairly robust for the specific set of agents and environments. The article is clearly written.

# Weaknesses

In general, I find the article lacking in its contextualisation with the exploration literature, and lacking in its presentation of results, and choice of environments to test. The authors follow Eberhard 2023 very closely, including the choice of environments to test. However, many of those environments are ones in which exploration is not particularly useful. This is evidenced by the highly overlapping curves in figures 2 & 3. While comparison to Eberhard 2023 is good, this work could and should have extended to other domains, which wouldn't have been hard to do, presumably, nor cost significantly more compute.

The authors _must_ make sure they distinguish their contribution from Eberhard 2023. For example, Figure 1 is lifted verbatim from the published work without attribution.

Table 1 is very hard to navigate, and it is the main result of the work. The motivation for doing oracle, anti and gain only for mean performance is never justified, and error intervals are not given for _any_ reward results. As a consequence, it is hard to evaluate the qualitative discussion of results. Also, the majority of the work is just replicating Eberhard 2023, and has very little scientific import.

The authors claim that schedulers are better than pink noise, but they don't explain why. Particularly as this contradicts Eberhard 2023. Is there any particular reason to trust this work more than previous work? Is there an error here or in the previous article? Would this translate to other domains? This is a missed opportunity to address a novel finding.

The definition of the noise schedules is lacking. The precise mappings are critical if the authors are to claim that the speed of decay of a noise schedule is crucial to performance. Obviously there has to be some inherent temporal scaling if one is to use a cosine as a schedule, for example, but this is never clarified. Yet the authors claim that `atanh` is too strong a decay, and arbitrarily set a starting point of 2.5 instead of 2.0. The validity of this is impossible to evaluate without knowing the exact parameterisation of the schedule function.

The authors claim that a cosine schedule in noise is the most consistent, but this seems puzzling considering Figure 2 doesn't show this clearly: cosine is better in pendulum, but that is the noisiest env; cosine is the worst in walker-run. If anything, linear seems more consistent. But what does consistent mean anyway? This is not quantified, and intuitive assessment is expected.

And on this topic of intuitive judgments. I think the article would benefit from quantifying its claims better. How is it that authors declare spatio-temporal noise better than pink noise? Is it just eye-balling a table of results without confidence intervals nor a clear winner? That's not enough, unfortunately.

Finally, the article should be better supported and contrasted to literature.

---

> ### Author Response · Authors · 2024-05-07
> **Reply to Reviewer cEPF**
>
> We appreciate your detailed feedback and insightful comments.
>
> We would first like to clarify that our work is primarily focused on reproducing the findings of Eberhard et al. (2023). We felt like the original paper somewhat disregarded noise schedulers citing pink noise as the default choice for action noise. In our work, we propose two more candidates for action noise - noise schedulers and spatio-temporal noise.
>
> >This is evidenced by the highly overlapping curves in Figures 2 & 3.
>
> We obtain the shaded regions by using the worst value among our runs from each run at each time step as the lower bound and the highest value among our runs at each time step as the upper bound. This, coupled with the fact that adjusting the action noise distribution does not warrant a significant increase in expected reward over pink noise, explains the overlaps between the graphs. This is because pink noise at least performs decently for every choice of environment that we have previously experimented on.
>
> We do not expect to observe substantial differences in results between the three scheduler types, as they inherently follow a similar approach: they start with an Ornstein-Uhlenbeck (OU) noise-like distribution and gradually shift towards a white noise distribution over time. A small adjustment to the action noise distribution should not significantly impact the returns.
>
> Furthermore, we consider any noise distribution that can at least match the performance of pink noise across all environments and provide higher returns in some environments to be a better result. Our intention is to present pink noise, noise schedulers, and the proposed spatio-temporal noise as viable candidates for action noise distributions, depending on the characteristics of the environment.
>
> We acknowledge that to statistically conclude which noise distribution is a better candidate as a general action noise, considerably more experimentation would be required, which is challenging for us to undertake given our current resources.
>
> >The authors follow Eberhard 2023 very closely, including the choice of environments to test. However, many of those environments are ones in which exploration is not particularly useful.
>
> The environment selection was prioritized to align with Eberhard et al. (2023), as their work served as the foundation for our reproducibility study. Our primary objective was to validate their findings within the same set of environments.
>
> >For example, Figure 1 is lifted verbatim from the published work without attribution.
>
> Regarding Figure 1, we would like to clarify that it was not directly lifted from Eberhard et al. (2023). Instead, it was reproduced using the code provided in their supplementary material, applied to a different seed for the bounded integrator environment. We apologize for any confusion this may have caused.
>
> >The authors claim that schedulers are better than pink noise, but they don't explain why. Particularly as this contradicts Eberhard 2023.
>
> As highlighted in our paper, our intuition suggests that the model should initially explore globally when encountering an unfamiliar environment. As the model gains familiarity with the environment, it should gradually shift towards more localized exploration. The various noise scheduler functions (color schedulers) were designed to model this intuition. One potential reason why the cosine scheduler performed better in the noisy pendulum environment could be that the agent required an extended period of exploration resembling Ornstein-Uhlenbeck noise to discover high-reward regions in the policy space. Once these regions were identified, the specific noise distribution employed might not have a significant impact.
>
> >The definition of the noise schedules is lacking. The precise mappings are critical if the authors are to claim that the speed of decay of a noise schedule is crucial to performance.
>
> We have provided precise equations for the colored noise family in the appendix. Additionally, we have included descriptions of the dynamics for the bounded integrator and oscillator environments in the appendix.
>
> >How is it that authors declare spatio-temporal noise better than pink noise?
>
> After some thought, we conclude that at max, spatio-temporal noise gives a minimal increase in returns compared to pink noise and that does not warrant the added complexity. We do not expect significantly better results by just adjusting the action noise distribution.
>
> We hope this clarifies our intentions and addresses the concerns raised in your feedback. Please let us know if you require any further clarification or have additional comments.
>
> Thank you for your time and valuable input, which has undoubtedly contributed to improving the quality of our work.

---

### Decision · Action_Editor_9eC9 · 2024-05-22

**Recommendation:** Reject

**Comment:**

This article investigates the impact of various policy noise types on exploration in continuous control tasks. The work aims to reproduce and extend the results presented in (Eberhard et al., 2023). Specifically, it investigates the effect of using white noise (independent Gaussian samples), red noise (Ornstein-Uhlenbeck noise), and pink noise (a hybrid of white and red noise). Additionally, the study evaluates schedules that transition between different noise types throughout the training process and proposes a new technique combining both spatial and temporal conditioning.

The paper completes a reproducibility study of the work in (Eberhard et al., 2023). While this is of value, all reviewers are in general agreement that the paper should provide additional insights. All three reviewers recommend rejecting the paper.  They agree that, in its current form, the experimental evidence presented is not sufficient to back the claims.

Reviewers wSqx and H45F consider that the experiments were ran with an insufficient number of random seeds, limiting the statistical significance of the results.

Reviewers wSqx considers that confidence reporting lacks robust statistical measures, relying on mean performance with error bars representing differences from min/max performance. In addition, the reviewer considers that strong claims are made despite overlapping confidence intervals, particularly in the sections with the proposed innovations over the work in (Eberhard et al., 2023), namely sections 3.3 (noise scheduling) and 3.4 (spatio-temporal noise).

Reviewer H45F and cEPF consider that the work would benefit from other ways of quantitatively comparing exploration, e.g. state space coverage. They also consider that the paper would benefit from considering more environments (expanding those used in (Eberhard et al., 2023).

The AE wants to thank the authors for their answers and the work they put in the revision (adding several clarifications and corrections). Unfortunately, this was not considered sufficient by the reviewers without improvements in the experimental evaluation as explained above.

The AE agrees with the recommendation made by the reviewers. The authors should consider incorporating the reviewers comments, and resubmit a major revision at a later time.

**Audience:**

The paper is appropriate for many readers of TMLR.

**Claims And Evidence:**

All three reviewers agree that, in its current form, the experimental evaluation presented in the paper does not support the main claims. The paper is an empirical study that aims at comparing different types of policy noise types on exploration in continuous control tasks. However, several shortcomings in the experimental results prevent cast doubts on the statistical validity of the results (e.g. small number of seeds, lack of robust statistical measures of confidence, small number of environments). Please refer to the main comment for a detailed description.

**Resubmission Of Major Revision:**

The authors may consider submitting a major revision at a later time.